# Introducing spatial availability, a singly-constrained measure of competitive accessibility

**Anastasia Soukhov**[1]*, **Antonio Páez**[1], **Christopher D. Higgins**[2], **Moataz Mohamed**[3]

**1** School of Earth, Environment and Society, McMaster University, Hamilton, ON, Canada, **2** Department of Geography & Planning, University of Toronto Scarborough, Toronto, ON, Canada, **3** Department of Civil Engineering, McMaster University, Hamilton, ON, Canada

* soukhoa@mcmaster.ca

**Data Availability Statement:** Data relevant to this study are available from Github at DOI:10.5281/zenodo.6450308 (https://soukhova.github.io/TTS2016R/) and the code at https://github.com/soukhova/Spatial-Availability-Measure.

## Abstract

Accessibility indicators are widely used in transportation, urban and healthcare planning, among many other applications. These measures are weighted sums of reachable opportunities from a given origin, conditional on the cost of movement, and are estimates of the potential for spatial interaction. Over time, various proposals have been forwarded to improve their interpretability: one of those methodological additions have been the introduction of competition. In this paper we focus on competition, but first demonstrate how a widely used measure of accessibility with congestion fails to properly match the opportunity-seeking population. We then propose an alternative formulation of accessibility with competition, a measure we call *spatial availability*. This measure relies on proportional allocation balancing factors (friction of distance and population competition) that are equivalent to imposing a single constraint on conventional gravity-based accessibility. In other words, the proportional allocation of opportunities results in a *spatially available opportunities* value which is assigned to each origin that, when all origin values are summed, equals the total number of opportunities in the region. We also demonstrate how Two-Stage Floating Catchment Area (2SFCA) methods are equivalent to spatial availability and can be reconceptualized as singly-constrained accessibility. To illustrate the application of spatial availability and compare it to other relevant measures, we use data from the 2016 Transportation Tomorrow Survey of the Greater Golden Horseshoe area in southern Ontario, Canada. Spatial availability is an important contribution since it clarifies the interpretation of accessibility with competition and paves the way for future applications in equity analysis (e.g., spatial mismatch, opportunity benchmarking, policy intervention scenario analysis).

## Introduction

The concept of accessibility in transportation studies derives its appeal from the combination of the spatial distribution of opportunities and the cost of reaching them [1, 2]. Accessibility analysis is employed in geography [3, 4], public health [5–9], real estate valuation [10], tourism [11], and transportation [12, 13] among other areas, with the number of applications growing

**Funding:** The lead author is supported by funding from the Social Sciences and Humanities Research Council's (SSHRC) partnership grant: Mobilizing justice: towards evidence-based transportation equity policy and the SSHRC Canadian Graduate Scholarship - Doctoral award. The funders had no role in study design, data collection and analysis, decision to publish, or preparation of the manuscript.

**Competing interests:** The authors have declared that no competing interests exist.

[14], especially as mobility-based planning is de-emphasized in favor of access-oriented planning [15–18].

Accessibility analysis stems from the foundational works of Harris (1954) [19] and Hansen (1959) [1]. From these seminal efforts, many accessibility measures have been derived, particularly after the influential work of Wilson (1971) [20] on spatial interaction (nb., utility-based measures derive from a very different theoretical framework, random utility maximization). Of these, gravity-type accessibility is arguably the most common; since its introduction in the literature, it has been widely adopted in numerous forms [13, 21–25]. Hansen-type accessibility indicators are essentially weighted sums of opportunities, with the weights given by an impedance function that depends on the cost of movement and thus measures the *intensity of the possibility* of interaction [1]. This type of accessibility analysis offers a powerful tool to study the intersection between urban structure and transportation infrastructure [2].

Despite their usefulness, the interpretability of Hansen-type accessibility measures can be challenging [13, 26]. Since they aggregate opportunities, the results are sensitive to the size of the region of interest (e.g., a large city has more jobs than a smaller city). As a consequence, raw outputs are not necessarily comparable across study areas [27]. This limitation becomes evident when surveying studies that implement this type of analysis. For example, studies [5] (in Montréal) and [28] (in Nairobi) report accessibility as the number of health care facilities that can potentially be reached from origins. But what does it mean for a zone to have accessibility to less than 100 facilities in each of these two cities, with their different populations and number of facilities? For that matter, what does it mean for a zone to have accessibility to more than 700 facilities in Montréal, besides being "accessibility rich"? As another example, studies [29] (in Bogota), [30] (in Montréal), and [31] (in Beijing) report accessibility as numbers of jobs, with accessibility values often in the hundreds of thousands, and even exceeding one million jobs for some zones in Beijing and Montréal. As indicators of urban structure, these measures are informative, but the meaning of one million accessible jobs is harder to pin down: how many jobs must any single person have access to? Clearly, the answer to this question depends on how many people demand jobs.

The interpretability of Hansen-type accessibility has been discussed in numerous studies, including recently by [32, 33], and in greater depth by [34]. As hinted above, the limitations in interpretability are frequently caused by ignoring competition; without competition, each opportunity is assumed to be equally available to every single opportunity-seeking individual that can reach it [33, 35, 36]. This assumption is appropriate when the opportunity of interest is non-exclusive, that is, if use by one unit of population does not preclude use by another. For instance, national parks with abundant space are seldom used to full capacity, so the presence of some population does not exclude use by others. When it comes to exclusive opportunities, or when operations may be affected by congestion, the solution has been to account for competition. Several efforts exist that do so. In our reckoning, the first such approach was proposed by Weibull (1976) [37], whereby the distance decay of the supply of employment and the demand for employment (by workers) were formulated under so-called axiomatic assumptions. This approach was then applied by Joseph (1984) [38] in the context of healthcare, to quantify the availability of general practitioners in Canada. About two decades later, Shen (1998) [35] independently re-discovered Weibull's [37] formula [see footnote (7) in Shen (1998 [35]] and deconstructed it to consider accessibility for different modes. These advances were subsequently popularized as the family of Two-Stage Floating Catchment area (2SFCA) methods [39] that have found widespread adoption in healthcare, education, and food systems [40–44].

An important development contained in Shen (1998)'s work is a proof that the population-weighted sum of the accessibility measure with competition equates to the number of opportunities available [footnote (7) and Appendix A in 35]. This demonstration gives the impression that Shen-type accessibility allocates *all* opportunities to the origins, however to our knowledge, it has not been interpreted by literature in this way. For instance, [34, 45, 46] all use Shen-type accessibility to calculate job access but report values as 'competitive accessibility scores' or simply 'job accessibility'. These works do not explicitly recognize that jobs that are assigned to each origin are in fact a proportion of *all* the opportunities in the system. This recognition, we argue, is critical to interpreting the meaning of the final result. Thus, in this paper we intend to revisit accessibility with competition within the context of disentangling how opportunities are allocated. We first argue that Shen (1998)'s competitive accessibility ('Shen-type accessibility' in this paper) misleadingly refers to the total zonal population to equal the travel-cost discounted opportunity-seeking population. This equivocation, we believe, results in an ambiguous interpretation of what Shen-type accessibility represents as the allocation of opportunities to population is masked by the results presenting as rates (i.e., opportunities per capita). We then propose an alternative formulation of accessibility that incorporates competition by adopting a proportional allocation mechanism; we name this measure *spatial availability*. The use of balancing factors for proportional allocation is akin to imposing a single constraint on the accessibility indicator, in the spirit of Wilson (1971)'s [20] spatial interaction model.

The key motivations of this paper are as follows:

- To address and improve on the interpretability of Hansen-type accessibility measure; and

- To consider competition from the perspective of the population for opportunities within an accessibility measure.

In this way, the paper's aim is three-fold:

- First, we aim to demonstrate that Shen-type (and hence Weibull (1976) [37] accessibility and the popular 2SFCA method) produce equivocal estimates of opportunities allocated as the result is presented as a rate (i.e., opportunities per capita);

- Second, we introduce a new measure, *spatial availability*, which we submit is a more interpretable alternative to Shen-type accessibility, since opportunities in the system are preserved and proportionally allocated to the population; and

- Third, we show how Shen-type accessibility (and 2SFCA methods) can be seen as measures of singly-constrained accessibility.

Discussion is supported by the use of the small synthetic example of Shen (1998) [35] and empirical data drawn from the 2016 Transportation Tomorrow Survey of the Greater Toronto and Hamilton Area in Ontario, Canada. In the spirit of openness of research in the spatial sciences [47–49] this paper has a companion open data product [50]. All code is available for replicability and reproducibility purposes at the cited GitHub repository [51].

## Accessibility measures revisited

In this section we revisit Hansen-type and Shen-type accessibility indicators. We adopt the convention of using a capital letter for absolute values (number of opportunities) and lower case for rates (opportunities per capita).

## Hansen-type accessibility

Hansen-type accessibility measures follow the general formulation shown in Eq (1):

$$S_i = \sum_{j=1}^{J} O_j \cdot f(c_{ij}) \qquad (1)$$

where:

- $c_{ij}$ is a measure of the cost of moving between $i$ and $j$.

- $f(\cdot)$ is an impedance function of $c_{ij}$; it can take the form of any monotonically decreasing function chosen based on positive or normative criteria [52].

- $i$ is a set of origin locations ($i = 1, \cdots, N$).

- $j$ is a set of destination locations ($j = 1, \cdots, J$).

- $O_j$ is the number of opportunities at location $j$; $O = \sum_{j=1}^{J} O_j$ is the total supply of opportunities in the study region.

- $S$ is Hansen-type accessibility as weighted sum of opportunities.

As formally defined, accessibility $S_i$ is the sum of opportunities that can be reached from location $i$, weighted down by an impedance function of the cost of travel $c_{ij}$. Summing the opportunities in the neighborhood of $i$ provides estimates of the number of opportunities that can *potentially* be reached from $i$. Several measures result from using a variety of impedance functions; for example, cumulative opportunities measures are obtained when $f(\cdot)$ is a binary or indicator function [13, e.g., 30, 53, 54]. Other measures use impedance functions modeled after any monotonically decreasing function [e.g., Gaussian, inverse power, negative exponential, or log-normal, among others, see, *inter alia*, 55–58]. In practice, accessibility measures with different impedance functions tend to be highly correlated [55, 59, 60].

Gravity-based accessibility has been shown to be an excellent indicator of the intersection between spatially distributed opportunities and transportation infrastructure [14, 55, 57]. However, beyond enabling comparisons of relative values, they are not highly interpretable on their own [26]. To address the issue of interpretability, previous research has aimed to index and normalize values on a per demand-population basis [61–63]. However, as recent research on accessibility discusses [27, 33, 34, 36], these steps do not adequately consider competition. In effect, when calculating $S_i$, every opportunity enters the weighted sum once for every origin $i$ that can reach it. This makes interpretability opaque, and to complicate matters, can also bias the estimated landscape of opportunity.

## Shen-type competitive accessibility

To account for competition, the influential works of Shen (1998) [35] and Weibull (1976) [37], as well as the widely used 2SFCA approach of Luo & Wang (2003) [39], adjust Hansen-type accessibility to account for the population's demand for opportunities in the region of interest. The mechanics of this approach consist of calculating, for every destination $j$, the population that can reach the opportunity(ies) given the impedance function $f(\cdot)$; let us call this the *effective opportunity-seeking population* (Eq (2)). This value can be seen as the Hansen-type *market area* (accessibility to population) of $j$. The opportunities at $j$ are divided by the sum of the effective opportunity-seeking population to obtain a measure of opportunities per capita, i.e., $R_j$ in Eq (3). This can be thought of as the *level of service* at $j$. Per capita values are then allocated back to the population at $i$, again subject to the impedance function as seen in Eq (4); this is

accessibility with competition.

$$P_{ij}^* = P_i \cdot f(c_{ij}) \tag{2}$$

$$R_j = \frac{O_j}{\sum_i P_{ij}^*} \tag{3}$$

$$a_i = \sum_j R_j \cdot f(c_{ij}) \tag{4}$$

where:

- $a$ is Shen-type accessibility as weighted sum of opportunities per capita (or weighted level of service).

- $c_{ij}$ is a measure of the cost of moving between $i$ and $j$.

- $f(\cdot)$ is an impedance function of $c_{ij}$.

- $i$ is a set of origin locations ($i = 1, \cdots, N$).

- $j$ is a set of destination locations ($j = 1, \cdots, J$).

- $O_j$ is the number of opportunities at location $j$; $O = \sum_{j=1}^{J} O_j$ is the total supply of opportunities in the study region.

- $P_i$ is the population at location $i$.

- $P_{ij}^*$ is the population at location $i$ that can reach destination $j$ according to the impedance function; we call this the *effective opportunity-seeking population*.

- $R_j$ is the ratio of opportunities at $j$ to the sum over all origins of the *effective opportunity-seeking population* that can reach $j$; in other words, this is the total number of opportunities per capita found at $j$.

Shen describes $P_i$ as the *"the number of people in location i seeking opportunities"* [35]. In our view, this is somewhat equivocal and where misinterpretation of the final results may arise. Consider a population center where the population is only willing to take an opportunity if the trip required is less than or equal to 60 minutes. This travel behaviour is captured by the following impedance function:

$$f(c_{ij}) = \begin{cases} 1 \text{ if } c_{ij} \leq 60 \text{ min} \\ 0 \text{ otherwise} \end{cases} \tag{5}$$

If an employment center is less than 60 minutes away, the population can seek opportunities there (i.e., $f(c_{ij}) = 1$). But are these people still part of the opportunity-seeking population for jobs located two hours away? How about four hours away? We assume that they are not part of the opportunity-seeking population because their travel behaviour, as represented by the impedance function, would yield $f(c_{ij}) = 0$, eliminating them from the effective opportunity-seeking population $P_{ij}^*$. We see Shen's definition as ambiguous because, for the purpose of calculating accessibility, the impedance function defines what constitutes the population that effectively can seek opportunities at remote locations. Thus, $P_i$ should be plainly understood as the population at location $i$ (as defined above) and not the *"the number of people in location i seeking opportunities"*. In other words, $P_i$ and $P_{ij}^*$ are confounded.

Furthermore, an identical misunderstanding can be described for $O_j$ which is defined as *"the number of **relevant** opportunities in location j"* in [35] (our emphasis). $O_j$ is adjusted by the same $f(c_{ij})$ in Eq (4), so the *relevancy* is determined by the travel behaviour associated with the impedance function and not only by $O_j$. For this reason, $O_j$ should be understood plainly as the opportunities at location $j$ (as defined above).

Misunderstanding $P_i$ and $O_j$ may lead to a misleading interpretation of the final result $a_i$, especially as expressed in Shen's proof (see Eq (6)).

$$\sum_{i=1}^{N} a_i P_i = \sum_{j=1}^{J} O_j \tag{6}$$

Confounding $P_i$ with the effective opportunity-seeking population and confounding $O_j$ with the jobs taken may cause us to misunderstand $a_i$ as *"relevant opportunities"* per *"people in location i seeking opportunities"*. Instead, as mathematically expressed in Shen's proof, $a_i$ is a proportion of the opportunities available to the population, since multiplying $a_i$ by the population at $i$ and summing for all origins in the system equals to the total number of opportunities in the system. Embedded in $a_i$ is already the travel behaviour, so $P_i$ and $O_j$ must be plainly understand as the population at $i$ and opportunities at $j$ for Eq (6) to hold true.

## Shen's synthetic example

In this section we use the synthetic example in Shen (1998) [35] to highlight the importance of understanding $P_i$ and $O_j$ as simply the population at the origin $i$ and the opportunities at destination $j$, respectively. This is critical to understanding how opportunities are allocated to the population based on the impedance function.

Table 1 contains the information needed to calculate $S_i$ and $a_i$ for this example. We use a negative exponential impedance function with $\beta = 0.1$ as also used in Shen (1998) [35, see

**Table 1. Summary description of the synthetic example: Hansen-type accessibility $S_i$, Shen-type accessibility $a_i$, and spatial availability $V_i$ with beta = 0.1 (light yellow) and beta = 0.6 (light grey).**

| Origin | A | | | B | | | C | | |
|---|---|---|---|---|---|---|---|---|---|
| Dest. | 1 | 2 | 3 | 1 | 2 | 3 | 1 | 2 | 3 |
| Pop. | 50000 | 50000 | 50000 | 150000 | 150000 | 150000 | 10000 | 10000 | 10000 |
| Jobs | 100000 | 100000 | 10000 | 100000 | 100000 | 10000 | 100000 | 100000 | 10000 |
| TT | 15 | 30 | 100 | 30 | 15 | 100 | 100 | 100 | 15 |
| f(TT) | 0.223 | 0.050 | < 0.001 | 0.050 | 0.223 | < 0.001 | < 0.001 | < 0.001 | 0.223 |
| Pop * f(TT) | 11156.5 | 2489.4 | 2.3 | 7468.1 | 33469.5 | 6.8 | 0.5 | 0.5 | 2231.3 |
| Jobs * f(TT) | 22313.0 | 4978.7 | 0.5 | 4978.7 | 22313.0 | 0.5 | 4.5 | 4.5 | 2231.3 |
| S_i | 27292.2 | 27292.2 | 27292.2 | 27292.2 | 27292.2 | 27292.2 | 2240.4 | 2240.4 | 2240.4 |
| a_i | 1.337 | 1.337 | 1.337 | 0.888 | 0.888 | 0.888 | 0.996 | 0.996 | 0.996 |
| f(TT) | < 0.001 | < 0.001 | < 0.001 | < 0.001 | < 0.001 | < 0.001 | < 0.001 | < 0.001 | < 0.001 |
| Pop * f(TT) | 6.170 | < 0.001 | < 0.001 | 0.002 | 18.511 | < 0.001 | < 0.001 | < 0.001 | 1.234 |
| Jobs * f(TT) | 12.341 | 0.002 | < 0.001 | 0.002 | 12.341 | < 0.001 | < 0.001 | < 0.001 | 1.234 |
| S_i | 12.343 | 12.343 | 12.343 | 12.343 | 12.343 | 12.343 | 1.234 | 1.234 | 1.234 |
| a_i | 1.999 | 1.999 | 1.999 | 0.667 | 0.667 | 0.667 | 1.000 | 1.000 | 1.000 |
| F^c | 0.238 | 0.238 | 0.238 | 0.714 | 0.714 | 0.714 | 0.048 | 0.048 | 0.048 |
| F^p | 0.817 | 0.182 | < 0.001 | 0.182 | 0.817 | < 0.001 | < 0.001 | < 0.001 | 1.000 |
| F | 0.599 | 0.069 | 0.001 | 0.401 | 0.931 | 0.003 | < 0.001 | < 0.001 | 0.996 |
| V_ij | 59900.6 | 6922.7 | 10.1 | 40096.9 | 93076.0 | 30.4 | 2.4 | 1.3 | 9959.5 |
| V_i | 66833.5 | 66833.5 | 66833.5 | 133203.4 | 133203.4 | 133203.4 | 9963.2 | 9963.2 | 9963.2 |

footnote (5)]:

$$f(c_{ij}) = \exp(-\beta \cdot c_{ij})$$

In Table 1, we see that population centers $A$ and $B$ have equal Hansen-type accessibility ($S_A = S_B =$ 27,292 jobs). On the other hand, the isolated satellite town of $C$ has low accessibility ($S_C =$ 2,240 jobs). But center $B$, despite its high accessibility, is a large population center. $C$, in contrast, is smaller but also relatively isolated and has a balanced ratio of jobs (10,0000 jobs) to population (10,000 people). It is difficult from these outputs to determine whether accessibility at $C$ is better or worse than that at $A$ or $B$.

The results are easier to interpret when we consider Shen-type accessibility. The results indicate that $a_A \approx$ 1.337 jobs per capita, $a_B \approx$ 0.888, and $a_C \approx$ 0.996. The latter value is sensible given the jobs-population balance of $C$. Center $A$ is relatively close to a large number of jobs (more jobs than the population of $A$). The opposite is true of $B$. According to Shen (1998) [35], the sum of the population-weighted accessibility $a_i$ is exactly equal to the number of jobs in the region following Shen's proof:

$$\sum_{i=1}^{N} a_i P_i = \sum_{j=1}^{J} O_j$$

$$50,000 \times 1.3366693$$

$$+150,000 \times 0.8880224$$

$$+10,000 \times 0.9963171 = 210,000$$

As mentioned earlier, this property under Shen's definition of $P_i$ "*people in location i seeking opportunities*", gives the impression that all jobs sought are allocated to the people located at each origin $i$. In other words, Shen defines $P_i$ to mean $P_{ij}^*$ (i.e., the *effective opportunity-seeking population* that is already adjusted by travel behaviour) instead of defining it to simply be the full population at $i$ (i.e., $P_i$). As seen in column **Pop** $*$ **f(TT)** in Table 1 (i.e., $P_{ij}^* = P_i \cdot f(c_{ij})$), the number of individuals from population center $A$ that are *willing to reach* employment centers 1, 2, and 3 are 11,156, 2,489, and 2.27 respectively. Therefore, the total effective opportunity-seeking population at $A$ is $P_A^* = \sum_j P_{Aj}^*$, that is, 13,647.27 people: this is considerably lower than the total population of $A$ (i.e., $P_A =$ 50,000 people). Demonstrated as follows, using $P_{ij}^*$ in the calculations associated with this proof results in only 56,834.59 jobs being allocated to the population, instead of the nominal number of jobs in the region that is over three times this number (i.e., 210,000 jobs).

$$\sum_{i=1}^{N} a_i P_{ij}^* =$$

$$(11,156.51 + 2,489.35 + 2.26) \times 1.3366693$$

$$+(7,468.06 + 33,469.52 + 6.81) \times 0.8880224$$

$$+(4.54 + 4.54 + 2,231.20) \times 0.9963171 \approx 56,834.59$$

Furthermore, even when Shen's $P_i$ is understood plainly as the total population at $i$, the meaning of the proof may still be ambiguous. The proof can still give the impression that all jobs are allocated to the total population, since total population ($\sum_{i=1}^{N} P_i$) goes into the equation and total jobs ($\sum_{j=1}^{J} O_j$) in the region is the result. However, this impression is incomplete as it does not reflect the magnitude of population that takes jobs and the number of people

being considered for jobs: these magnitudes are a product of the impedance function. The magnitudes are not obvious because the result, $a_i$, is a rate (i.e., opportunities per capita).

Let us consider a modification to the travel behaviour of the example discussed to illustrate how the presentation of $a_i$ as a rate obscures the magnitude of the effective opportunity-seeking population. We modify the example by increasing the $\beta$ to 0.6 (compared to the previous value of 0.1; see Fig 1). This modification increases the cost of travel and thus the impedance function (an expression of the population's relative willingness to travel to opportunities). The modification reflects a population that is relatively less willing to travel to opportunities further away compared to the previous $\beta$ value. The Hansen-type and Shen-type values are presented in the yellow rows of Table 1.

As expected, Hansen-type accessibility drops quite dramatically after this $\beta$ modification: the friction of distance is so high that few opportunities are within reach. In contrast, Shen-type accessibility converges to the jobs:population ratio (i.e., origin $A$ is $\frac{100,000}{50,000} = 2$). This is explained by the way the impedance function excludes the population in droves, thus reducing the competition for jobs: as seen in Table 1, the effective opportunity-seeking population from $A$ is only about equal to 6.17. Likewise, the number of jobs at 1 weighted by the impedance is only 12.341. In other words, competition is low (limited onto the the population close to job centers) because jobs are expensive to reach, but those willing to reach jobs enjoy relatively high accessibility (in the limit, the jobs:population ratio). On the other hand, the accessibility is effectively zero for those in the population prevented by the impedance from reaching any jobs (the population not near the job centers).

In what follows, we propose an alternative derivation of Shen (1998) [35] accessibility with competition that explicitly clarifies the opportunities allocated to the *effective opportunity-seeking population* within its formulation. Hence, the results are not only more interpretable, but also extend the potential of accessibility analysis.

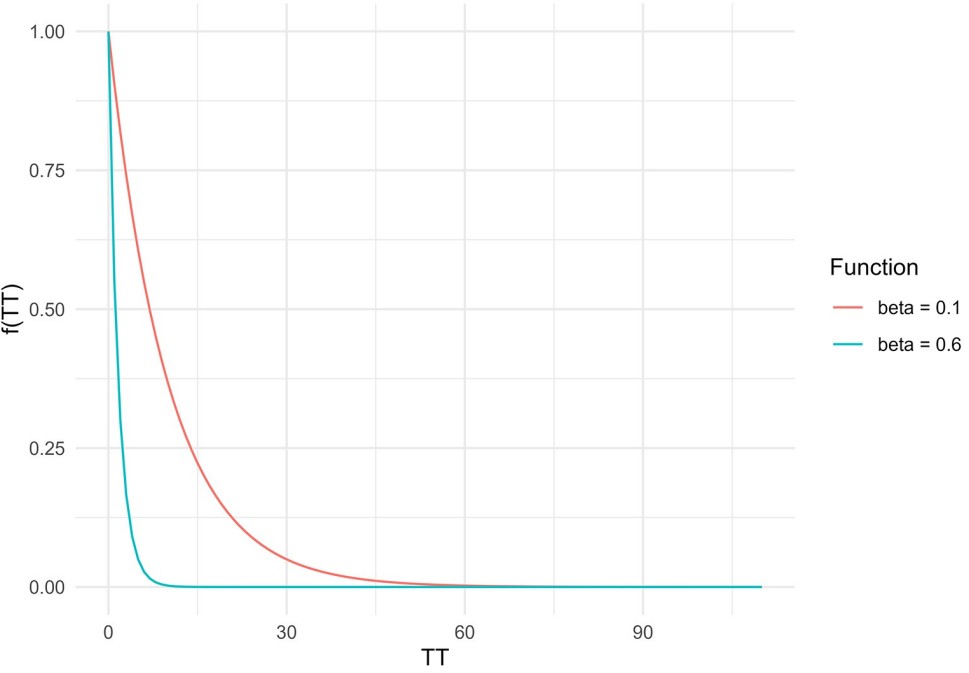

**Fig 1. Comparison of two negative exponential impedance functions used in the synthetic example.** The x-axis represents the travel time (in mins) and the y-axis represents the impedance function at each travel time.

## Introducing spatial availability: A singly-constrained measure of accessibility

In brief, we define the *spatial availability* at $i$ ($V_i$) as the proportion of all opportunities $O$ that are allocated to $i$ from all destinations $j$:

$$V_i = \sum_{j=1}^{N} O_j F_{ij}^t$$

where:

- $F_{ij}^t$ is a balancing factor that depends on the population and cost of movement in the system.

- $O_j$ is the number of opportunities at $j$.

- $V_i$ is the number of spatially available opportunities from the perspective of $i$.

The general form of spatial availability is also as a sum, and the fundamental difference between Hansen- and Shen-type accessibility is that opportunities are allocated proportionally. Balancing factor $F_{ij}^t$ consists of two components: a population-based balancing factor $F_i^p$ and an impedance-based balancing factor $F_{ij}^c$ that, respectively, allocate opportunities to $i$ in proportion to the size of the population of the different competing centers (the mass effect of the gravity model) and the cost of reaching opportunities (the impedance effect). In the next two subsections, we explain the intuition behind the method before defining it in full.

### Proportional allocation by population

According to the gravity modelling framework, the potential for interaction depends on the mass (i.e., the population) and the friction of distance (i.e., the impedance function). We begin by describing the proposed proportional allocation mechanism based on demand by the population. Recall, the total population in the example is 210,000. The proportion of the population by population center is:

$$F_A^p = \frac{50,000}{210,000}$$

$$F_B^p = \frac{150,000}{210,000}$$

$$F_C^p = \frac{10,000}{210,000}$$

Jobs are allocated proportionally from each employment center to each population center depending on their population sizes as per the balancing factors $F_i^p$. In this way, employment center 1 allocates $100,000 \cdot \frac{50,000}{210,000} = 23,809.52$ jobs to $A$; $100,000 \cdot \frac{150,000}{210,000} = 71,428.57$ jobs to $B$; and $100,000 \cdot \frac{10,000}{210,000} = 7,142.857$ jobs to $C$. Notice how this mechanism ensures that the total number of jobs at employment center 1 is preserved at 100,000.

We can verify that the number of jobs allocated is consistent with the total number of jobs in the system:

Employment center 1 to population centers A, B, and C :

$$100,000 \cdot \frac{50,000}{210,000} + 100,000 \cdot \frac{150,000}{210,000} + 100,000 \cdot \frac{10,000}{210,000} = 100,000$$

Employment center 2 to population centers A, B, and C :

$$100,000 \cdot \frac{50,000}{210,000} + 100,000 \cdot \frac{150,000}{210,000} + 100,000 \cdot \frac{10,000}{210,000} = 100,000$$

Employment center 3 to population centers A, B, and C :

$$10,000 \cdot \frac{50,000}{210,000} + 10,000 \cdot \frac{150,000}{210,000} + 10,000 \cdot \frac{10,000}{210,000} = 10,000$$

In the general case where there are $N$ population centers in the region, we define the following population-based balancing factors in Eq (7):

$$F_i^p = \frac{P_i^\alpha}{\sum_{i=1}^{N} P_i^\alpha} \tag{7}$$

Balancing factor $F_i^p$ corresponds to the proportion of the population in origin $i$ relative to the population in the region. On the right hand side of the equation, the numerator $P_i^\alpha$ is the population at origin $i$. The summation in the denominator is over $i = 1, \cdots, N$, and adds up to the total population of the region. Notice that we incorporate an empirical parameter $\alpha$. The role of $\alpha$ is to modulate the effect of demand by population. When $\alpha < 1$, opportunities are allocated more rapidly to smaller centers relative to larger ones; $\alpha > 1$ achieves the opposite effect.

Balancing factor $F_i^p$ can now be used to proportionally allocate a share of available jobs at $j$ to origin $i$. The number of jobs available to $i$ from $j$ balanced by population shares is defined as follows:

$$V_{ij}^p = O_j \frac{F_i^p}{\sum_{i=1}^{N} F_i^p}$$

In the general case where there are $J$ employment centers, the total number of jobs available from all destinations to $i$ is simply the sum of $V_{ij}^p$ over $j = 1, \cdots, J$:

$$V_i^p = \sum_{j=1}^{J} O_j \frac{F_i^p}{\sum_{i=1}^{N} F_i^p}$$

Since the factor $F_i^p$, when summed over $i = 1, \cdots, N$ always equals to 1 (i.e., $\sum_{i=1}^{N} F_i^p = 1$), the sum of all spatially available jobs equals $O$, the total number of opportunities in the region:

$$\sum_{i=1}^{N} V_i^p = \sum_{i=1}^{N} \sum_{j=1}^{J} O_j \frac{F_i^p}{\sum_{i=1}^{N} F_i^p}$$

$$= \sum_{i=1}^{N} \frac{F_i^p}{\sum_{i=1}^{N} F_i^p} \cdot \sum_{j=1}^{J} O_j$$

$$= \sum_{j=1}^{J} O_j = O$$

The terms $F_i^p$ act here as the balancing factors of the gravity model when a single constraint is imposed (i.e., to ensure that the sums of columns are equal to the number of opportunities per destination, see Ortúzar & Willumsen (2011) [64], pp. 179–180 and pp. 183–184). As a result, the sum of spatial availability for all population centers equals the total number of opportunities.

The discussion so far concerns only the mass effect (i.e., population size) of the gravity model. In addition, the potential for interaction is thought to decrease with increasing cost, so next we define similar balancing factors but based on the impedance.

## Proportional allocation by cost

Clearly, using only balancing factor $F_i^p$ to calculate spatial availability $V_i^p$ does not account for the cost of reaching employment centers. Consider instead a set of balancing factors $F_{ij}^c$ that account for the friction of distance for our example. Recall that the impedance function $f(c_{ij})$ equals $\exp(-\beta \cdot c_{ij})$ where $\beta = 0.1$ and travel time $c_{ij}$ is either 15, 30 or 60 minutes. For instance, the impedance-based balancing factors $F_{ij}^c$ would be the following for employment center 1 (employment center 2 and 3 have their own balancing factor values for each origin $i$ as will be discussed later):

$$F_{A1}^c = \frac{0.223130}{0.223130 + 0.049787 + 0.000045} = 0.8174398$$

$$F_{B1}^c = \frac{0.049787}{0.223130 + 0.049787 + 0.000045} = 0.1823954$$

$$F_{C1}^c = \frac{0.000045}{0.223130 + 0.049787 + 0.000045} = 0.0001648581$$

Balancing factors $F_{ij}^c$ use the impedance function to proportionally allocate more jobs to closer population centers, that is, to those with populations *more willing to reach the jobs*. Indeed, the factors $F_{ij}^c$ can be thought of as the proportion of the population at $i$ willing to travel to destination $j$, conditional on the travel behavior as given by the impedance function. For instance, 81.74398% of jobs from employment center 1 are allocated to population center $A$ based on impedance.

So as follows from our example, of the 100,000 jobs at employment center 1 the number of jobs allocated to population center $A$ is $100,000 \times 0.8174398 = 81,743.98$ jobs; the number allocated to population center $B$ is $100,000 \times 0.1823954 = 18,239.54$ jobs; and the number allocated to population center $C$ is $100,000 \times 0.0001648581 = 16.48581$ jobs. We see once more that the total number of jobs at the employment center is preserved at 100,000. In this example, the proportional allocation mechanism assigns the largest share of jobs to population

center $A$, which is the closest to employment center 1 and the smallest to the more distant population center $C$.

In the general case where there are $N$ population centers and $J$ employment centers in the region, we define the following impedance-based balancing factors:

$$F_{ij}^c = \frac{f(c_{ij})}{\sum_{i=1}^{N} f(c_{ij})} \tag{8}$$

The total number of jobs available to $i$ from $j$ according to impedance is defined as follows:

$$V_{ij}^c = O_j \frac{F_{ij}^c}{\sum_{i=1}^{N} F_{ij}^c}$$

The total number of jobs available to $i$ from all destinations is:

$$V_i^c = \sum_{j=1}^{J} O_j \frac{F_{ij}^c}{\sum_{i=1}^{N} F_{ij}^c}$$

Like the population-based allocation factors, $F_i^c$ summed over $i = 1, \cdots, N$ always equals to 1 (i.e., $\sum_{i=1}^{N} F_{ij}^c = 1$). As before, the sum of all spatially available jobs equals $O$, the total number of opportunities in the region:

$$\sum_{i=1}^{N} V_i^c = \sum_{i=1}^{N} \sum_{j=1}^{J} O_j \frac{F_{ij}^c}{\sum_{i=1}^{N} F_{ij}^c}$$

$$= \sum_{i=1}^{N} \frac{F_{ij}^c}{\sum_{i=1}^{N} F_{ij}^c} \cdot \sum_{j=1}^{J} O_j$$

$$= \sum_{j=1}^{J} O_j = O$$

We are now ready to more formally define spatial availability with due consideration to both population and travel cost effects.

## Assembling mass and impedance effects

Population and the cost of travel are both part of the gravity modelling framework. Since the balancing factors defined in the preceding sections are proportions, they can be combined multiplicatively to obtain their joint effect. This multiplicative relationship can alternatively be understood as the joint probability of allocating opportunities and is captured by Eq (9), where $F_i^p$ is the population-based balancing factor that grants a larger share of the existing opportunities to larger centers and $F_{ij}^c$ is the impedance-based balancing factor that grants a larger share of the existing opportunities to closer centers. This is in line with the tradition of gravity modeling.

$$F_{ij}^t = \frac{F_i^p \cdot F_{ij}^c}{\sum_{i=1}^{N} F_i^p \cdot F_{ij}^c} \tag{9}$$

with $F_i^p$ and $F_{ij}^c$ as defined in Eqs (7) and (8) respectively. The combined balancing factor $F_{ij}^t$ is

used to proportionally allocate jobs from $j$ to $i$. Hence, spatial availability is given by Eq (10).

$$V_i = \sum_{j=1}^{J} O_j \, F_{ij}^t \tag{10}$$

The terms in Eq (10):

- $F_{ij}^t$ is a balancing factor as defined in Eq (9).

- $i$ is a set of origin locations in the region $i = 1, \cdots, N$.

- $j$ is a set of destination locations in the region $j = 1, \cdots, J$.

- $O_j$ is the number of opportunities at location $j$.

- $V_i$ is the spatial availability at $i$.

Notice that, unlike $S_i$ in Hansen-type accessibility (Eq (1)), the population enters the calculation of $V_i$ through $F_i^p$. Returning to the example in Fig 2, Table 1 also contains the information needed to calculate $V_i$, with $\beta$ set again to 0.1. Column **V_ij** are the jobs available to each origin from each employment center. In this column $V_{A1} = 59{,}901$ is the number of jobs

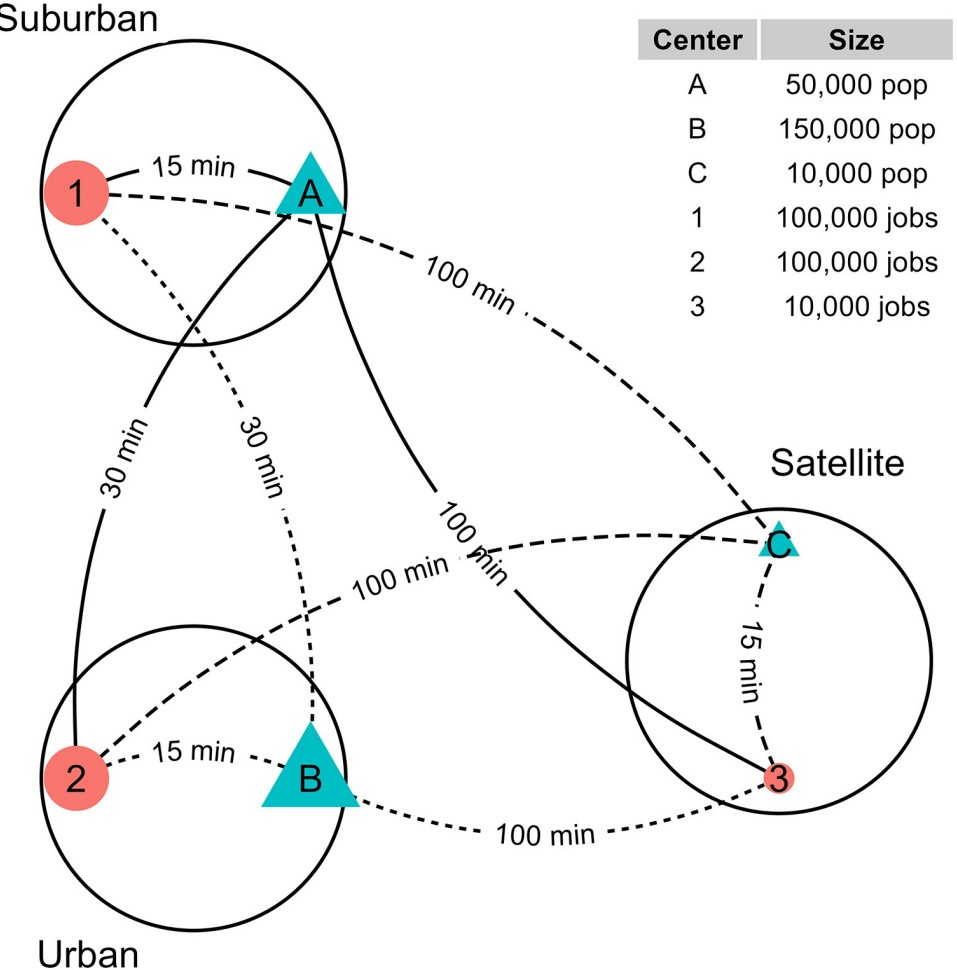

| Center | Size |
|--------|---------------|
| A | 50,000 pop |
| B | 150,000 pop |
| C | 10,000 pop |
| 1 | 100,000 jobs |
| 2 | 100,000 jobs |
| 3 | 10,000 jobs |

**Fig 2. Shen (1998) synthetic example with locations of employment centers (in orange), population centers (in blue), number of jobs and population and travel times.**

available at $A$ from employment center 1. Column $\mathbf{V\_i}$ (i.e., $\sum_{j=1}^{J} V_{ij}$) gives the total number of jobs available to origin $i$. We can verify that the total number of jobs available is consistent with the total number of jobs in the region (with some small rounding error):

$$\sum_{i=1}^{N} V_i = 66,833 + 133,203 + 9,963 \approx 210,000$$

Compared the calculated values of $V_i$ to column $\mathbf{S\_i}$ (Hansen-type accessibility) in Table 1. The spatial availability values are more intuitive. Recall that population centers $A$ and $B$ had identical Hansen-type accessibility to employment opportunities. According to $V_i$, population center $A$ has greater job availability due to: 1) its close proximity to employment center 1; combined with 2) less competition (i.e., a majority of the population have to travel longer distances to reach employment center 1). Job availability is lower for population center $B$ due to much higher competition (150,000 people can reach 100,000 jobs at equal cost). And center $C$ has almost as many jobs available as it has population.

As discussed above, Hansen-type accessibility is not designed to preserve the number of jobs in the region. Shen-type accessibility ends up preserving the number of jobs in the region but the definitions of variables are internally obscured; the only way it preserves the number of jobs is if the effect of the impedance function is ignored when expanding the values of jobs per capita to obtain the total number of opportunities. The proportional allocation procedure described above, in contrast, consistently returns a number of jobs available that matches the total number of jobs in the region.

Since the jobs spatially available are consistent with the jobs in the region, it is possible to define a measure of spatial availability per capita as presented in Eq (11):

$$v_i = \frac{V_i}{P_i} \tag{11}$$

And, since the jobs are preserved, it is possible to use the regional jobs per capita ($\frac{\sum_{j=1}^{J} O_j}{\sum_{i=1}^{N} P_i}$) as a benchmark to compare the spatial availability of jobs per capita at each origin.

In the example, since the population is equal to the number of jobs, the regional value of jobs per capita is 1.0. To complete the illustrative example, the spatial availability of jobs per capita by origin is:

$$v_1 = \frac{V_1}{P_1} = \frac{66,833.47}{50,000} = 1.337$$

$$v_2 = \frac{V_2}{P_2} = \frac{133,203.4}{150,000} = 0.888 \tag{12}$$

$$v_3 = \frac{V_3}{P_3} = \frac{9,963.171}{10,000} = 0.996$$

We can see that population center $A$ has fewer jobs per capita than the regional benchmark, center $B$ has more, and center $C$ is at parity. Remarkably, the spatial availability per capita matches the values of $a_i$ in Table 1. S1 Appendix has a proof of the mathematical equivalence between the two measures. It is interesting to notice how Weibull (1976) [37] and Shen (1998) [35], as well as this paper, all reach identical expressions starting from different assumptions; this effect is known as *equifinality* (see Ortúzar & Willumsen (2011) [64], p. 333 and Williams

(1981) [65]). This result means that Shen-type accessibility and 2SFCA can be re-conceptualized as singly-constrained accessibility measures along with the proposed spatial availability measure.

## Why does proportional allocation matter?

We have shown that Shen-type accessibility and spatial availability produce equifinal results when accessibility per-capita is computed. At this point it is reasonable to ask whether the distinction between these two measures is of any importance.

Conceptually, we would argue that the confounded populations in Shen-type accessibility leads to internal inconsistency in the calculation of total opportunities in Shen (1998) [35]: this points to a deeper issue that is only evident when we consider the intermediate values of the method. To illustrate, Table 1 shows results of $a_i$ that are reasonable (and they match exactly the spatial availability per capita). But when we dig deeper, these results mask potentially misleading values for the jobs allocated and the number of jobs taken. For instance, a region with a high jobs:population ratio but a prohibitive transportation network that results in a high cost of travel may yield a high $a_i$ value. This value, however, can conceal a low *effective opportunity-seeking population* and a proportionally low number of allocated jobs, while also obscuring the magnitude of the population that does and does *not* take jobs.

In addition, the intermediate accessibility values of $a_i$ (Shen-type measure) may also lead to impact estimates that are deceptive [66]. For example, the estimated region-wide cost of travel considering the jobs allocated by $a_i$ in Table 1 (i.e., $Jobs^*f(TT)$) is as follows:

$$22,313 \times 15 \text{ min} + 4,979 \times 30 \text{ min} + 0.454 \times 100 \text{ min} +$$

$$4,979 \times 30 \text{ min} + 22,313 \times 15 \text{ min} + 0.454 \times 100 \text{ min} +$$

$$4.54 \times 100 \text{ min} + 4.54 \times 100 \text{ min} + 2,231 \times 15 \text{ min} = 1,002,594 \text{ min}$$

In contrast, the estimated region-wide cost of travel according to $V_i$ in Table 1 is as follows:

$$59,901 \times 15 \text{ min} + 6,923 \times 30 \text{ min} + 10 \times 100 \text{ min} +$$

$$40,097 \times 30 \text{ min} + 93,076 \times 15 \text{ min} + 30 \times 100 \text{ min} +$$

$$2.4 \times 100 \text{ min} + 1.3 \times 100 \text{ min} + 9,959 \times 15 \text{ min} = 3,859,054 \text{ min}$$

Often referred to as 'the supply of jobs' (or simply Hansen-type accessibility) in the Shen-type measure: $Jobs^*f(TT)$ cannot be used to understand the region-wide cost of travel. Recall how we define $Pop^*f(TT)$ as the *effective opportunity-seeking population* ($P_{ij}^*$), $Jobs^*f(TT)$ similarly represents the *effective **opportunities allocated*** and sums to approximately 56,824 out of a total of 210,000 jobs. Like $Pop^*f(TT)$, the *effective opportunities allocated* to each origin is only a reflection of the impedance function and not the *actual* number of opportunities allocated to each origin. Therefore, the resulting 1,002,594 min is not a meaningful measure of the cost of travel in the system.

However, since spatial availability allocates the *actual* number of opportunities to each origin; the 3,859,054 min can be used to quantify the system-wide impacts of competitive accessibility in this region. We know spatial availability's output is the number of opportunities at each $i$ since the combined balancing factors allocate a proportional amount of the total opportunities to each $i$ such that the number of opportunities allocated to each $i$ sum to equal the total opportunities in the region.

## Empirical example of Toronto

In this section we illustrate the application of spatial availability through an empirical example. We use full-time employment origin-destination flows from the city of Toronto. Toronto is within the Greater Golden Horseshoe (GGH) (31,500 km²), the most densely populated and industrialized region in Canada. Within the GGH is the Greater Toronto and Hamilton (GTHA); it forms the most populous metropolitan regions in Canada and the core urban agglomeration in the GGH.

The GTHA contains the city of Toronto, the most populous city in Canada, and the focus of this empirical example; it is used to demonstrate the application of the proposed spatial availability measure along with how it compares to Hansen- and Shen-type measures. We explain the data and then detail the calculated comparisons.

### GGH data

We obtained full-time employment flows from the 2016 Transportation Tomorrow Survey (TTS). This household travel survey collects representative travel journey information from all 20 municipalities contained within the GGH area (Fig 3) every five years [67]. The data set includes origin to destination flows associated with full-time employed people at the level of Traffic Analysis Zones (TAZ) (3,764 TAZ in the GGH) i.e., the number of full-time workers at origin household TAZ (3,446,957 workers in the GGH) and their matching employment destination TAZ (3,081,900 associated jobs in the GGH). TAZ are a unit of spatial analysis that are defined as part of the TTS. More broadly, TAZ are commonly used to ascribe production and attraction of trips in the context of transportation planning modelling. In the GGH data set, TAZ contain on average 916 workers and 819 jobs (descriptive statistics in Fig 3). The TTS data is based on a representative sample of between 3% to 5% of households in the GGH and is weighted to reflect the population covering the study area as a whole [67].

To generate the travel cost for the full-time employment trips, travel times between origins and destinations (i.e., centroids of the TAZ) are calculated for car travel using the R package

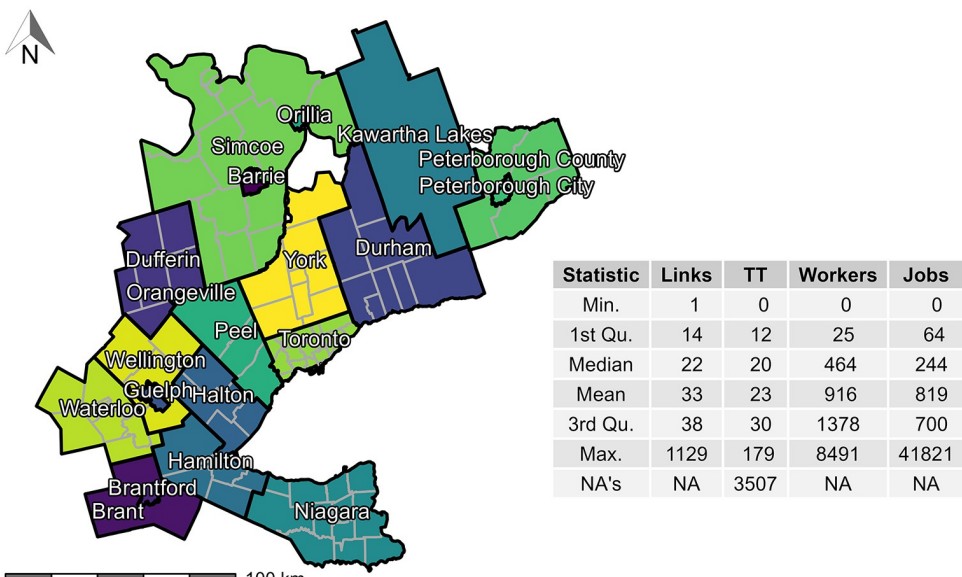

| Statistic | Links | TT | Workers | Jobs |
|---|---|---|---|---|
| Min. | 1 | 0 | 0 | 0 |
| 1st Qu. | 14 | 12 | 25 | 64 |
| Median | 22 | 20 | 464 | 244 |
| Mean | 33 | 23 | 916 | 819 |
| 3rd Qu. | 38 | 30 | 1378 | 700 |
| Max. | 1129 | 179 | 8491 | 41821 |
| NA's | NA | 3507 | NA | NA |

**Fig 3. TTS 2016 study area (GGH, Ontario, Canada) along with the descriptive statistics of the origin destination (OD) pairs (count of workers that travel to their place of employment) by origin TAZ, calculated OD car travel time (TT), workers per TAZ, and jobs per TAZ.** Contains 20 municipalities/regions (black boundaries) and subregions (dark gray boundaries).

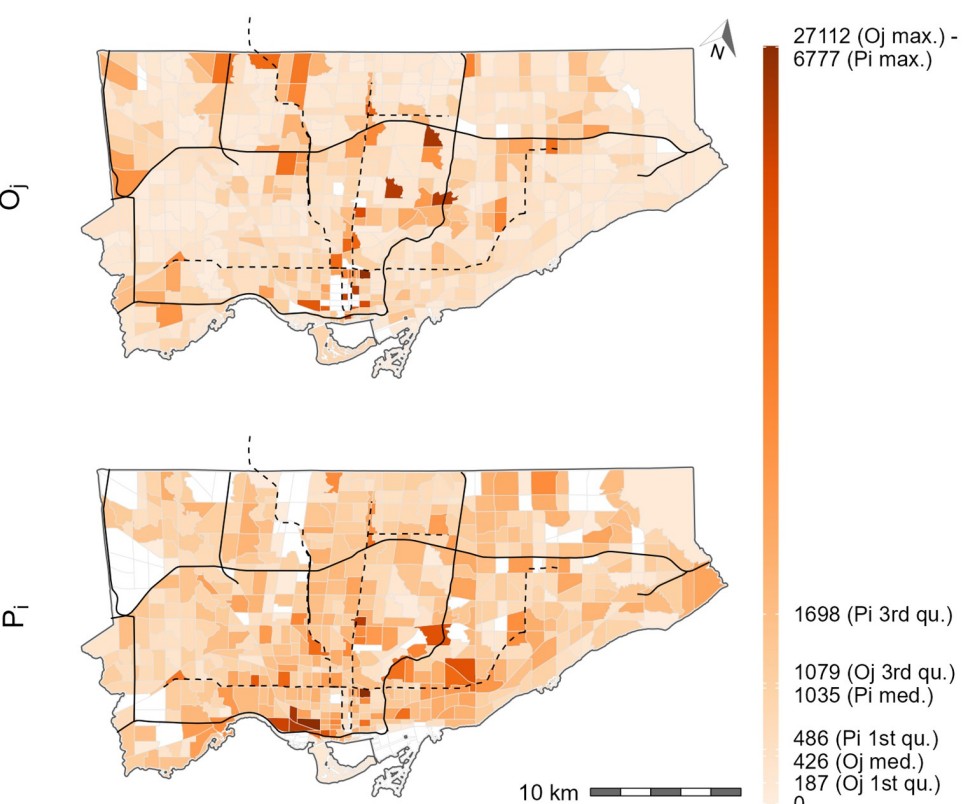

**Fig 4. Spatial distribution of full-time jobs (top) and full-time working population (bottom) at each TAZ for Toronto as provided by the 2016 TTS.** Black lines represent expressways and black dashed lines represent subway lines. All white TAZ have no worker population or jobs.

{r5r} with a street network retrieved from OpenStreetMap [68, 69]. For inter-TAZ trips, a 3 hr travel time threshold is selected as it captures 99% of population-employment pairs (travel times summarized in Fig 3). This method does not account for traffic congestion or modal split, which can be approximated through additional methods [70, 71]. For simplicity, we carry on with the assumption that all trips are taken by car in uncongested travel conditions. Additionally, we assume all intra-TAZ trips are equal to 0.1 minutes for illustrative purposes. All data and data preparation steps are documented and can be freely explored in the companion open data product {TTS2016R}.

## Spatial employment characteristics in Toronto

As mentioned, the focus of this empirical example is on the city of Toronto. It is the largest city in the GGH and represents a significant subset of workers and jobs in the GGH; 22% of workers in the GGH live in Toronto and 25% of jobs that these workers take are located within Toronto. The spatial distribution of jobs and workers is shown in Fig 4. A high density of jobs can be found in the central southern part of Toronto (the downtown core). Spatial trends in the distribution of workers is more uniform relative to the distribution of jobs.

Next, the spatial distribution of the estimated car travel time (green) and the associated standard deviation (grey) is visualized in Fig 5. Overall, it can be seen that car travel time is lower within the downtown core and higher outside of the downtown core. Trends from both plots indicate that trips originating from within the center of Toronto are shorter and more similar in length than trips originating from closer to the Toronto boundaries.

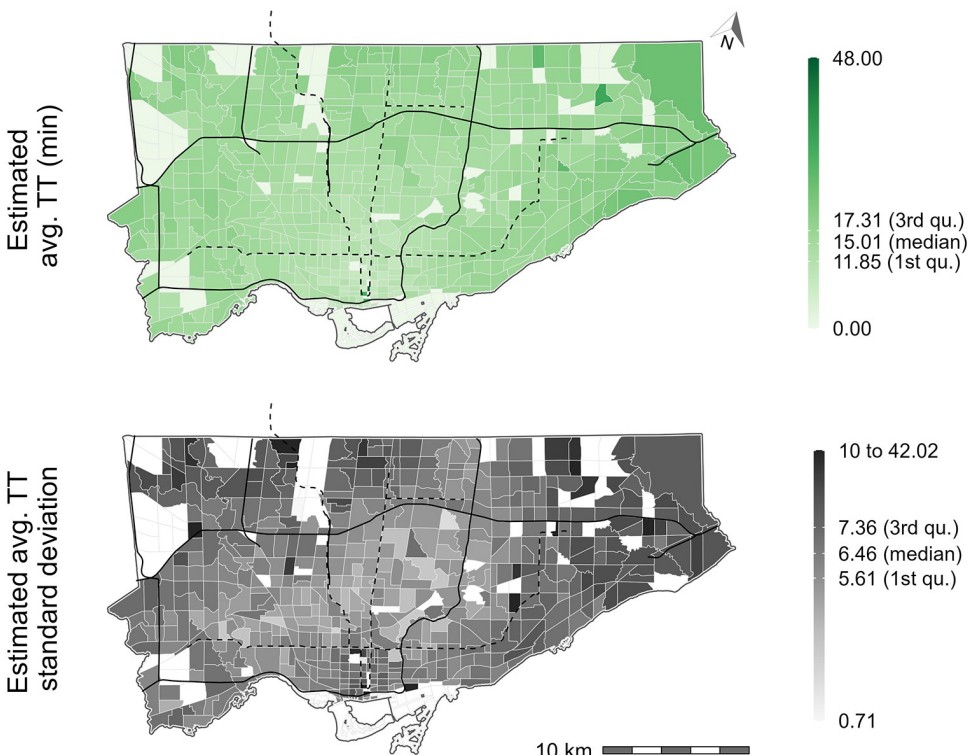

**Fig 5. Mean car travel time to jobs per origin (top) and standard deviation of car travel times per origin (bottom) for the city of Toronto.** Origin destination flows are provided by the 2016 TTS and car travel times estimated using r5r. Black lines represent expressways and black dashed lines represent subway lines. White TAZ represent a TAZ with no workers hence no travel time.

### Calibration of an impedance function for Toronto

In the synthetic example introduced in Section 2, we use a negative exponential function with the parameter reported by Shen (1998) [35]. For the empirical Toronto data set, we calibrate an impedance function on the trip length distribution (TLD) of commute trips. Briefly, a TLD represents the proportion of trips that are taken at a specific travel cost (e.g., travel time); this distribution is commonly used to derive impedance functions in accessibility research [72–74].

As mentioned, the calculations are undertaken for the city of Toronto using only the full-time employed population in the city and associated jobs in Toronto. Specifically, edge trips are not included, such as trips originating in Toronto but finishing outside of Toronto (i.e., jobs not in Toronto) and trips originating outside of Toronto but finishing in Toronto (i.e., full-time employed people living outside of Toronto). The empirical and theoretical TLD for this Toronto data set are represented in the top-left panel of Fig 6. Maximum likelihood estimation and the Nelder-Mead method for direct optimization available within the {fitdistrplus} package [75] is used. Based on goodness-of-fit criteria and associated diagnostics, the normal distribution is selected (see Fig 6).

For reference, the normal distribution is defined in Eq (13). It depends on a mean parameter $\mu$ (estimated to be 14.169) and a standard deviation parameter $\sigma$ (estimated to be 7.369).

$$f(x) = \frac{1}{\sigma\sqrt{2\pi}} e^{-\frac{1}{2}\left(\frac{x-\mu}{\sigma}\right)^2} \tag{13}$$

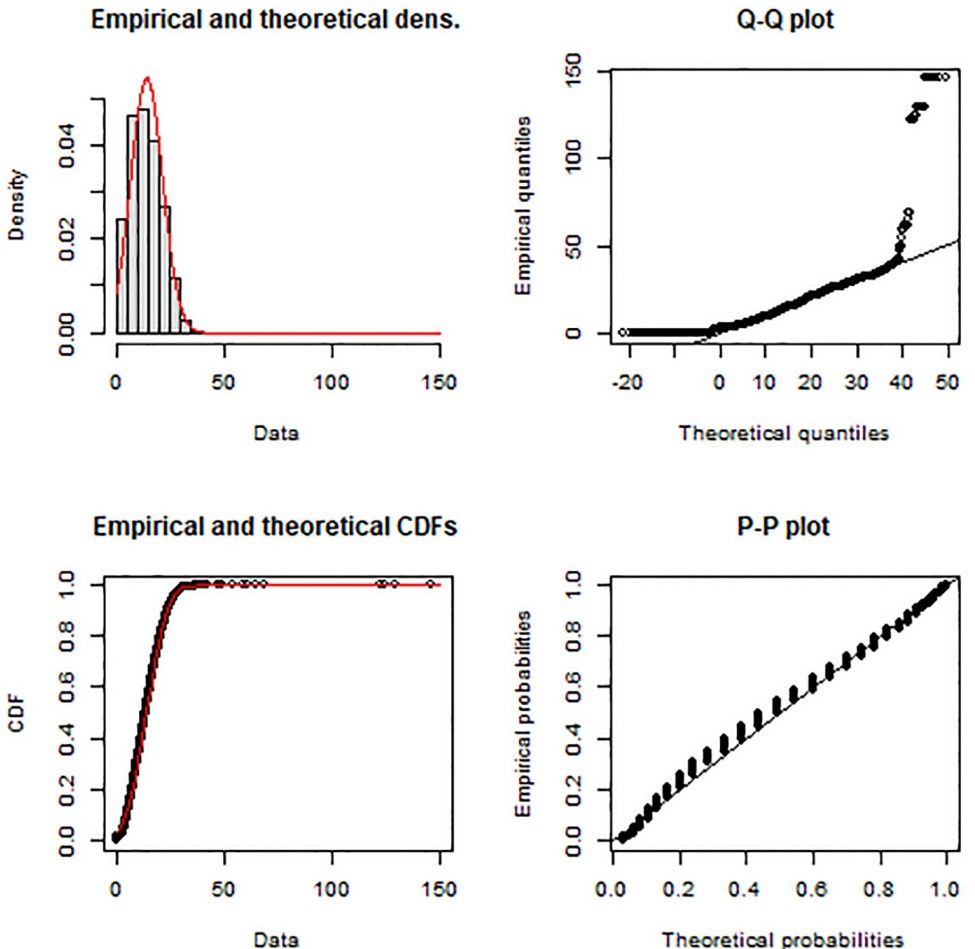

**Fig 6. Car trip length distribution and calibrated normal distribution impedance function (red line) with associated Q-Q and P-P plots.** Based on the estimated car travel times for full-time employment OD trips in Toronto from the TTS 2016.

Together, Figs 4 and 5 and the calibrated impedance function demonstrate the spatial distribution of worker and job data in Toronto and the inputs for this study's calculation of spatial availability, Shen- and Hansen- type measures.

### Accessibility and spatial availability of jobs in Toronto

**Absolute opportunity values.** Fig 7 contains the number of jobs accessible using Shen-type accessibility, Hansen-type accessibility, and the number of jobs *available* using the spatial availability measure. The values from all measures are represented on the same axis as they measure the absolute value of jobs accessible to the workers at the origin. In the top plot, the Shen-type accessibility is multiplied by the *effective opportunity-seeking population* to yield a value that corresponds to absolute number of accessible jobs (considering competition) according to Shen's definition. In the middle plot, the Hansen-type accessibility is an unconstrained case of accessibility where all jobs that are in-reach of each origin (according to the impedance function). In other words, in this plot, each value corresponds to the number of jobs that can be reach at each origin assuming no competition. Lastly, in the bottom plot, the proposed spatial availability measure is shown. Spatial availability is a constrained case of accessibility that proportionally allocates the number of jobs, at each origin, considering

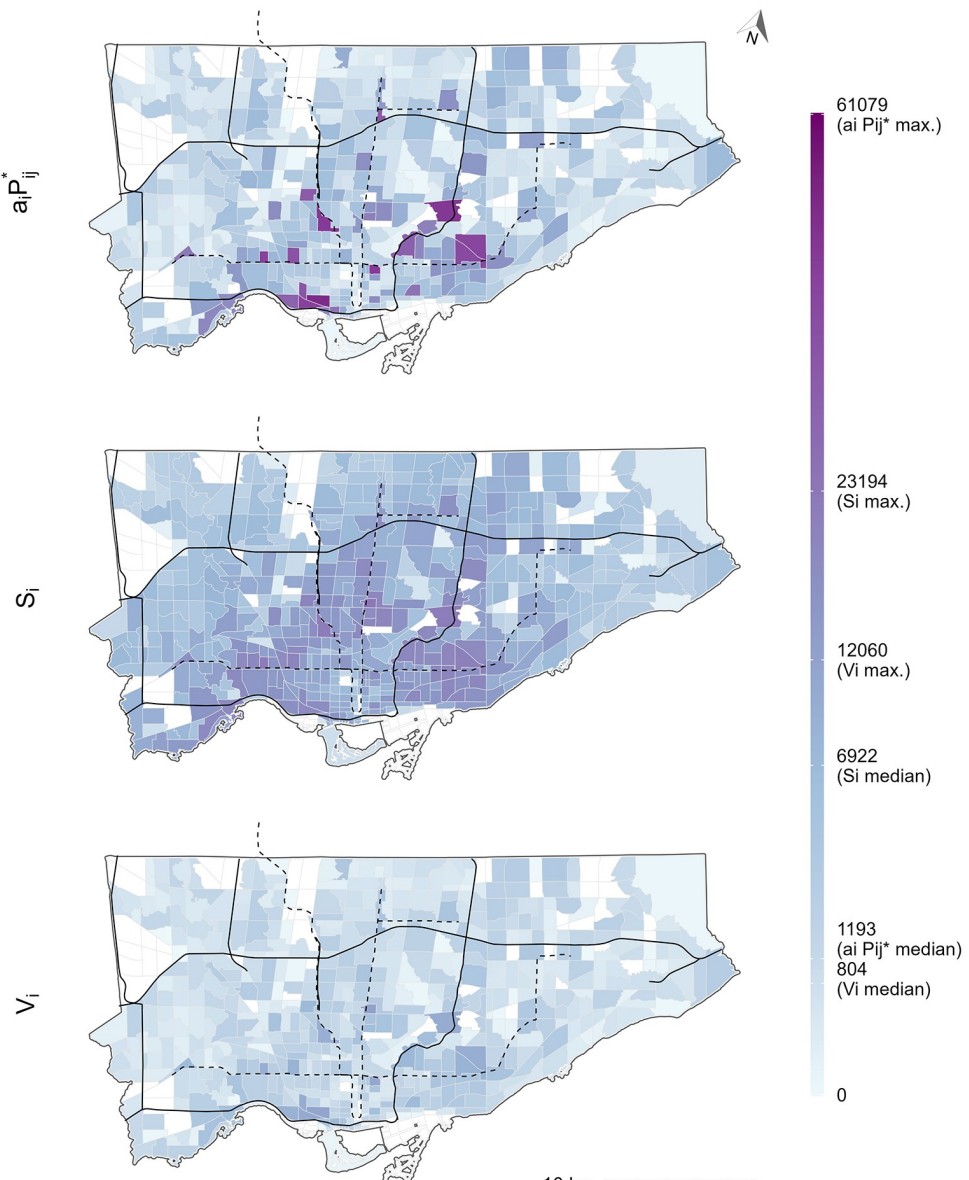

**Fig 7. Estimated accessibility to jobs (# of jobs) in Toronto according to Shen-type measure multipled by the effective opportunity-seeking population (top), Hansen-type measure (middle), and spatial availability (bottom).** Black lines represent expressways and black dashed lines represent subway lines. All white TAZ have no worker population or jobs, i.e., with null accessibility values. Legend scale is square root transformed to effectively visualize the range of values.

competition from the population in nearby origin and the relative travel cost (according to the impedance function).

What is notable about the bottom plot is that the proportional allocation mechanism of spatial availability ensures that the job availability value for each origin all sums to the city-wide total of 771,496 jobs (i.e., the number of destination flows from Toronto origins to Toronto destinations). The number of accessible jobs at each origin can therefore be interpreted as the number of *available* jobs to each origin based on the relative travel behaviour and density of competition for jobs (i.e., worker population). A proportion of each of the 771,496 jobs in Toronto are only allocated once to each origin. In terms of the middle plot, the city-wide total

sum of Hansen-type accessibility is 4,370,250 jobs. This value is relatively meaningless since the measure is unconstrained and represents the sum of opportunities that have been counted anywhere from 1 to many times depending on the impedance function. As previously discussed, unconstrained counting of the same opportunity by all origins is not an issue if the opportunity itself is non-exclusive, but since one job can only be given to one worker (especially since the worker and job data is derived from origin destination (OD) flows), it is inappropriate to use unconstrained measures to capture employment characteristics. Comparing the middle and bottom plots, it is evident that the unconstrained counting of opportunities (Hansen-type) results in absolute values that are higher throughout the city, particularly in TAZ that are in proximity to high job density (recall Fig 4). These same trends are not present in the spatial availability bottom plot as the absolute value is lower than Hansen-type accessibility middle plot since the proximity to high job density and competition from worker density is proportionally metered.

Lastly, the top plot visualizes the *absolute* Shen-type measure (as understood by Shen's definition of $P_i$ being equal to $P_{ij}^*$) and sums to the city-wide value of 2,125,281 jobs by multiplying $a_i$ by the *effective opportunity-seeking population* $P_{ij}^*$ (i.e., the denominator of the rate). This plot demonstrates how confounding $P_i$ with $P_{ij}^*$ yields an *incorrect* number of competitively accessible jobs: this is evidently incorrect because the sum of $a_i P_{ij}^*$ greatly exceeds the city-wide total of workers (i.e., 2,125,281 > 771,496). To the authors' knowledge, literature has not attempted to convert Shen-type accessibility to the absolute value of accessible jobs in the way demonstrated in the top plot: we suspect this is the case because of the ambiguous definition in the Shen-type measure that conflates $P_{ij}^*$ with $P_i$. If $a_i$ is multiplied by $P_i$, it yields the same value as $V_i$, but since the definition of Shen-type measure is equivocal, doing so is not clear since the denominator of $a_i$ (note: which is a rate) is *not* $P_i$. The resulting plot, spatially, is similar to spatial availability (bottom plot) but certain TAZ have exceptionally high values in an inconsistent way. This is because $a_i$ uses the impedance function values for both access to jobs (numerator) and the competition from neighboring workers (denominator $P_{ij}^*$) to adjust their impact: using $P_{ij}^*$ does not consistently isolate the absolute value of accessible jobs. However, readers should note that if $a_i$ is multiplied by $P_i$ it yields the same values at $V_i$ (bottom plot) (the proof for mathematically equivalency is in S1 Appendix). As also mentioned earlier, the formulation of the denominator and numerator of $a_i$ is ambiguous. To presume that multiplying it by $P_i$ would disentangle the rate and yield the absolute value of accessible and available (i.e., considering competition) jobs is not clear within Shen-type measure's definition. However, through the calculation of spatial availability $V_i$, the absolute value of accessible and available opportunities at a zone *i* is clear.

**Internal values.** Carrying on the discussion on how to retrieve the absolute value of *available* jobs using the Shen-type measure ($a_i$), Fig 8 highlights how differences between $P_i$ and $P_{ij}^*$ are not uniform across space; the values at each origin are equivalent to $\sum_j f(c_{ij})$. Recall, $P_i$ is the number of workers at each TAZ (city-wide sum of 771,496) while $P_{ij}^*$ is the number of workers who *seek* jobs (city-wide sum of 1,776,458) in that TAZ based on their travel behaviour. $P_{ij}^*$ is an internal value of $a_i$ and the top plot presents the ratio of $P_{ij}^*$ to $P_i$. It reflects how the effective opportunity-seeking population is sometimes inflated (i.e., impedance values is greater than 1) and others deflated (i.e., impedance value is less than 1) by the Shen-type measure ($a_i$). As such, using $P_{ij}^*$ to untangle the absolute job availability from $a_i$ instead of $P_i$ can lead to exaggerating the total travel time in the city since it does not represent the *actual* number of workers but the *effective* number of workers. For instance, when trying to calculate the city-wide travel time using $a_i P_{ij}^*$, Shen-type accessibility yields 501,114.9 [h] instead of an interpretable city-wide travel time of 183,802.4 [h] that results if spatial availability $V_i$ is used. Again, following

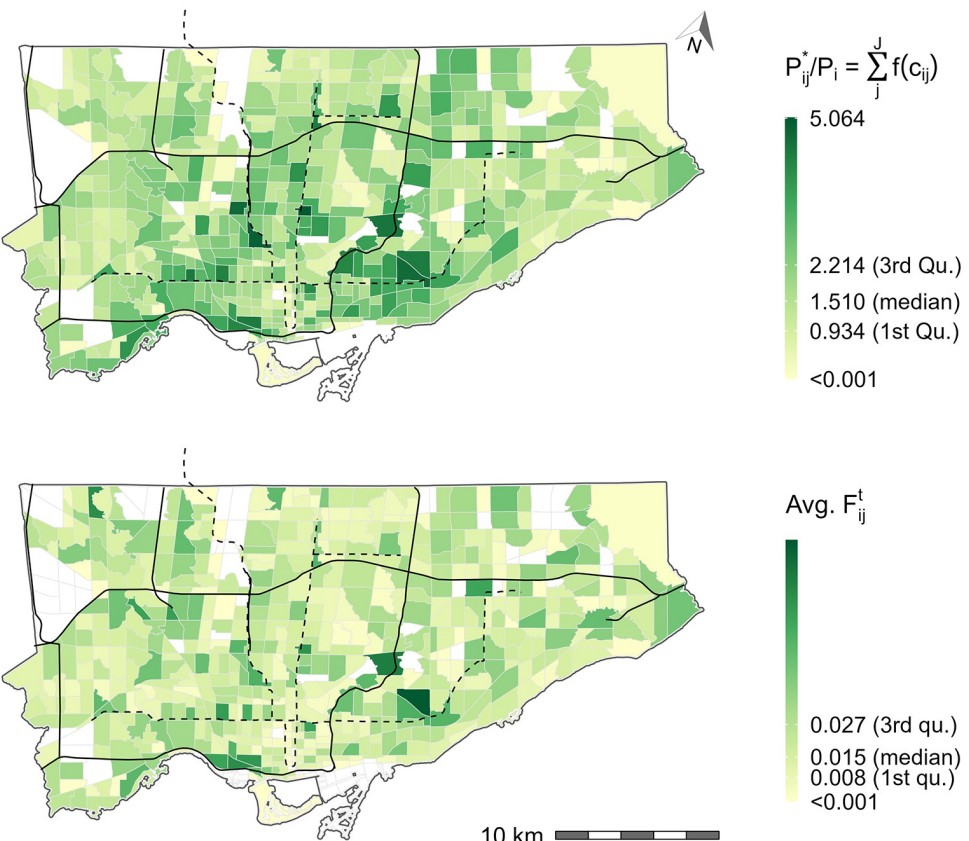

**Fig 8. The ratio of the effective opportunity seeking population to the population (top) and the average spatial availability's balancing factor (Eq (9)) (bottom) for Toronto TAZ.** Black lines represent expressways and black dashed lines represent subway lines. All white TAZ have no worker population or jobs, i.e., with null accessibility values.

the Shen-type accessibility definition, the absolute number of opportunities cannot be easily disentangled from $a_i$.

Not only are the number of available opportunities a direct result of $V_i$, the internal combined balancing factor $F_{ij}^t$ (Eq (9)) can be used for analysis. The bottom plot in Fig 8 demonstrates the average $F_{ij}^t$ for each TAZ. Practically, the visualized values corresponds to the average *proportion* of opportunities available that are claimed by the zone based on travel behaviour and population competition for opportunities. These values can allow the analyst to understand the magnitude of the *proportion of opportunities* that the origin TAZ is assigned based on the opportunities located at reachable destination TAZ. For instance, the TAZ with the maximum value of 0.090 has many origin to destination trips (112 trips, upper 3rd quantile), many workers (5,538 workers, upper 3rd quantile), and is located centrally within Toronto. Averaging $F_{ij}^t$ demonstrates that this TAZ claims on average a high proportion of jobs from reachable TAZ. This does not necessarily mean TAZ with a high $V_i$ have an exceptionally high average $F_{ij}^t$; many TAZ around the downtown core have high $V_i$ values but do not have exceptionally high average $F_{ij}^t$. The average $F_{ij}^t$ can therefore be used to identify relatively "greedy" areas that could possibly withstand reductions in availability if spatial availability in areas with a deficit of jobs available is increased (through policy intervention). The balancing factor is an interesting feature of spatial availability that opens avenues for future analysis; alas, there does not seem to be an equivalent within the current formulation of the Shen-type measure.

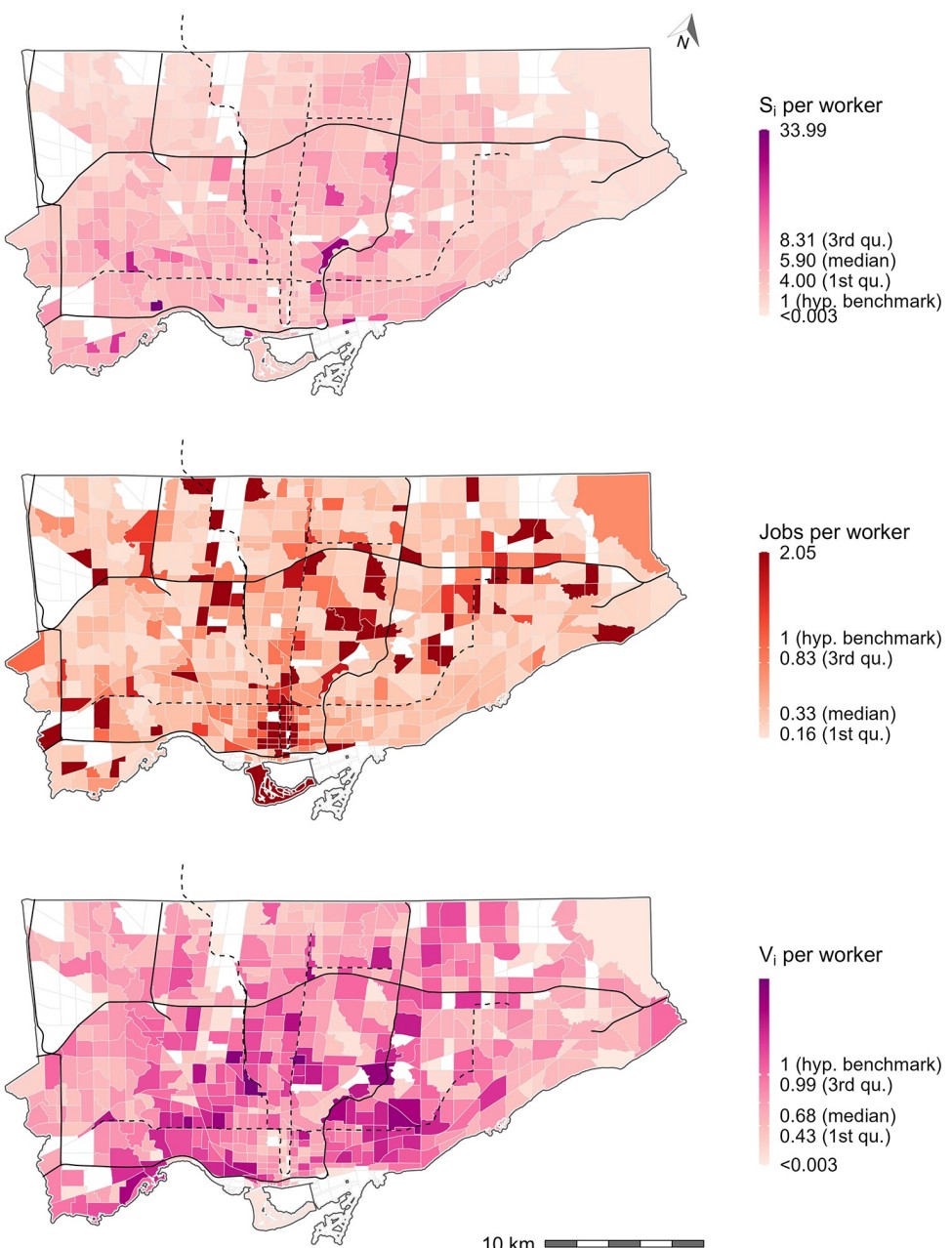

**Fig 9. Hansen-type accessible jobs per capita (top), number of jobs to population ratio (middle), and spatially available jobs per capita (bottom) for Toronto.** An arbitrary jobs-to-workers ratio is assumed to be 1 in the paper, this is the hypothetical opportunity availability benchmark. Black lines represent expressways and black dashed lines represent subway lines. All white TAZ have no worker population or jobs, i.e., with null accessibility values.

**Benchmarking opportunity availability.** Fig 9 presents the number of jobs per capita for Hansen-type accessibility (top plot), the raw number of jobs per capita (middle plot), and the spatially available jobs per capita (bottom plot). For the purpose of discussion, a hypothetical benchmark of 1 job per capita is assumed for the city.

The bottom plot displays the spatially available jobs per capita $v_i$. This value is mathematically equivalent to the Shen-type measure $a_i$, but with higher interpretability as a result of the

proportional allocation mechanism. This mechanism makes clear that all the opportunities are allocated proportionally to origins and $V_i$ values represent the number of available opportunities. $V_i$ can be directly divided by the population at the origin and expressed as opportunities per capita $v_i$. When spatial availability per capita (bottom plot) is compared to Hansen-type measure (top plot), dividing the output by population directly yields a more difficult to interpret number of *unconstrained* accessible jobs per capita. For instance, the median light-pink shaded TAZ corresponds to approximately 5.90 unconstrained accessible jobs per capita; this value is difficult to interpret because, as discussed in the Introduction Section, jobs are an exclusive opportunity types so their accessibility value should take into consideration competition.

The spatially available jobs per capita $v_i$ (bottom plot) can also be used for benchmarking. The values can be compared directly to the raw number of jobs per capita (middle plot) since the total number of opportunities are preserved (and the population, in this case, is equivalent to the number of opportunities). For instance, a TAZ with a $v_i > 1$ has more available jobs (based on travel behaviour and competition) than their working population. This TAZ has sufficient employment opportunities (under the assumptions of the input data), while a TAZ with a $v_i < 1$ does not have sufficient employment opportunities. From an equity perspective, $v_i$ can be used to target where residential housing, job opportunities, and/or transportation system improvements should be created.

For TAZ with $v_i$ values significantly greater than 1 (dark pinks), constructing more residential housing for the type of workers who occupy the *available jobs* in the proximate TAZ should be considered. Assuming accurate and realistic input data, increasing the competition in the area will decrease the $v_i$ score, but it can be decreased up to threshold of $v_i = 1$. For TAZ with $v_i$ values significantly less than 1 (light pinks), constructing more employment opportunities for the type of workers who live in proximate TAZ and/or prioritizing transportation network improvements to create more favourable travel time conditions can be considered.

Depending on the raw jobs per worker ratio, different approaches are appropriate. For instance, adding more residential locations near the downtown core (bottom center on the bottom plot) could be a good approach to increasing $v_i$ as there is already a high jobs per worker ratio (middle plot). However, doing so will decrease the $v_i$ availability in areas near the border of the city, so in addition, adding more employment opportunities to areas with low raw jobs per worker ratio and low $v_i$ is needed. Furthermore, the travel time landscape would also influence the resulting $v_i$ score, so transportation network improvements to areas with low $v_i$ could also be considered. This is to say, $v_i$ is dependent on the magnitude and spatial distribution of residential housing, job opportunities, and transportation system. The region could be optimized to achieve thresholds of specific $v_i$ values and thus the difference in residential housing, job opportunities, and transportation system can become policy intervention targets. It should also be kept in mind, that though $v_i = 1$ and the comparison to the raw jobs per worker values can be used for intervention planning, $v_i$ can easily be transformed back to $V_i$ to understand the magnitude of the job availability within that origin.

## Conclusion

In this paper we show how Shen-type accessibility, a widely used measure of accessibility with competition, obscures some important internal values associated with 'demanded' opportunities. The Shen-type measure's formulation confounds the population of origin-zones with the *effective opportunity-seeking population*. We propose an alternative derivation of accessibility with competition that we call *spatial availability* $V_i$. This measure ensures that opportunities are allocated in a proportional way and are preserved in the regional total. We demonstrate that spatial availability and Shen-type accessibility (along with 2SFCA) are equifinal: formally

the equations can be made equivalent and can be considered as singly-constrained (and competitive) accessibility measures, however, spatial availability has the additional feature of proportional allocation. We argue the feature of proportional allocation improves interpretability of the final values and, as follows, the per capita opportunity spatial availability values ($v_i$).

Why does spatial availability matter? Because competition in accessibility matters, so precise interpretation of the output is critical. Competition is an important consideration for certain opportunity types, and conventional Hansen-type accessibility does not capture it [34]. Through its intermediate values, spatial availability brings forward a different interpretation to competition than the Shen-type measure. As demonstrated in this paper, spatial availability increases interpretability by first presenting the absolute value of *available* jobs ($V_i$) and then by dividing the available jobs value by the number of working population ($v_i$). This rate $v_i$ is equivalent to Shen-type measure but contains internal values, such as the proportional allocation mechanism, that yield more realistic estimates of opportunities taken. Spatial availability's proportional allocation is a result of its balancing factors; these balancing factors can also be used to better understand the absolute and rate values obtained.

Spatial availability measure can be applied in many contexts, particularly in equity analysis and policy planning, fields where analysis of comparative and relative differences are pertinent. As shown in the empirical example, Hansen-type measure tends to result in values that are very extreme as a result of multiple-counting of opportunities. Multiple-counting may not be an issue if the opportunity-type is non-exclusive: but in the case of employment for example, where one worker can only take one job, the resulting values are difficult to interpret. In this respect, the increased interpretability and internal consistency of spatial availability can help to push accessibility analysis forward in ways Shen-type accessibility's formulation cannot. For instance, an analyst may use spatial availability if they are interested in analysing the internal values of their accessibility analysis (e.g., which zones are the most and least competitive in terms of travel cost and demanding-population as informed by the balancing factors). Additionally analysts may use spatial availability to understand or benchmark the magnitude of spatially available opportunities of zones under current transportation system conditions or within possible policy scenarios. Based on the research presented, we suggest the following guidelines for the application of spatial availability and topics of future work:

1. The Hansen-type accessibility should be used when opportunities are non-exclusive. When opportunities are exclusive (i.e., one opportunity spot for one person), the competitive accessibility measure of spatial availability should be strongly considered.

2. Shen-type accessibility can be used to compute the availability of jobs (the rate and the absolute values if the original definition is corrected), however, if the analyst is interested in internal values and secondary analysis of the results, spatial availability should be considered.

3. With the renewed interpretability of what the absolute *opportunity availability* is at each origin-zone, the spatial availability per capita $v_i$ value of 1 can be used as a policy goal: 1 opportunity per person. For areas with a value below 1, targeted increases to the quantity of opportunities, residential housing, and transportation system improvements can be considered such that the number of *available jobs* per capita in the zone is at least equal to 1. Since spatial availability per capita implicitly preserves the number of opportunities in the region, it can be directly compared to the region's raw jobs to population ratio to inform policy. Additionally, the absolute values of spatial availability can be used to understand the magnitude of the opportunity availability deficit/surplus.

4. Spatial availability per capita can also be compared directly to other regions as done by literature using Shen-type measure/2SFCA [9, 76, 77]. However, as a result of the renewed interpretation, the magnitude of *spatially available* opportunities can be quantified.

5. Lastly, since opportunities are preserved, many new avenues of analysis can be pursued. This is especially important in light of emerging concerns regarding accessibility inequities. For instance, the population and opportunities can be segmented (i.e., transit users, active transportation users, low income, low education, new comers, children) and their spatial availability to opportunities can be assessed, benchmarked, and corresponding policy to target inequities can be theorized. As another example, the combined balancing factor can be analysed to identify which populations currently do not seek opportunities because of friction of distance.

Lastly, please view the S1 Appendix for the mathematical equivalence of spatial availability per capita ($v_i$) and Shen-type accessibility ($a_i$). This document provides a step-by-step solution of $v_i$ for population center $A$ (from Shen's synthetic example [35] as discussed in Section 2.3).

## Supporting information

**S1 Appendix.**
(PDF)

## Author Contributions

**Conceptualization:** Anastasia Soukhov, Antonio Páez.

**Data curation:** Anastasia Soukhov, Antonio Páez.

**Formal analysis:** Anastasia Soukhov, Antonio Páez.

**Funding acquisition:** Anastasia Soukhov.

**Investigation:** Anastasia Soukhov, Antonio Páez.

**Methodology:** Anastasia Soukhov, Antonio Páez.

**Supervision:** Anastasia Soukhov.

**Validation:** Anastasia Soukhov, Antonio Páez, Moataz Mohamed.

**Visualization:** Anastasia Soukhov, Antonio Páez.

**Writing – original draft:** Anastasia Soukhov, Antonio Páez, Christopher D. Higgins.

**Writing – review & editing:** Anastasia Soukhov, Antonio Páez, Christopher D. Higgins, Moataz Mohamed.

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
