## [Decision Letter · Decision Letter 0]

12 Sep 2022

PONE-D-22-24454Introducing spatial availability, a singly-constrained measure of competitive accessibilityPLOS ONE

Dear Dr. Soukhov,

Thank you for submitting your manuscript to PLOS ONE. After careful consideration, we feel that it has merit but does not fully meet PLOS ONE’s publication criteria as it currently stands. Therefore, we invite you to submit a revised version of the manuscript that addresses the points raised during the review process.

We look forward to receiving your revised manuscript.

Kind regards,

Jun Yang

Academic Editor

PLOS ONE

Journal Requirements:

"The Social Sciences and Humanities Research Council’s partnership grant: Mobilizing justice:

towards evidence-based transportation equity policy."

4. We note that Figures 3,4,5,7,8 and 9 in your submission contain copyrighted images. All PLOS content is published under the Creative Commons Attribution License (CC BY 4.0), which means that the manuscript, images, and Supporting Information files will be freely available online, and any third party is permitted to access, download, copy, distribute, and use these materials in any way, even commercially, with proper attribution. For more information, see our copyright guidelines: http://journals.plos.org/plosone/s/licenses-and-copyright.

a. You may seek permission from the original copyright holder of Figures 3,4,5,7,8 and 9 to publish the content specifically under the CC BY 4.0 license. 

Additional Editor Comments:

Major Revision

Reviewers' comments:

Reviewer's Responses to Questions

**Comments to the Author**

1. Is the manuscript technically sound, and do the data support the conclusions?

Reviewer #1: Yes

Reviewer #2: Yes

2. Has the statistical analysis been performed appropriately and rigorously? 

Reviewer #1: Yes

Reviewer #2: Yes

3. Have the authors made all data underlying the findings in their manuscript fully available?

Reviewer #1: Yes

Reviewer #2: Yes

4. Is the manuscript presented in an intelligible fashion and written in standard English?

Reviewer #1: Yes

Reviewer #2: Yes

5. Review Comments to the Author

Reviewer #1: The authors demonstrated how a widely used measure of accessibility with congestion fails

to properly match the opportunity-seeking population.

Please highlight the key points in the abstract.

The introduction should further highlight the scientific problems, motivations, and possible innovations of the paper.

What is the contribution and why the contribution is important?

The results should be better described, discussed and justified. I suggest the results should be better discussed and justified, such as whether they are consistent with previous studies or analyzing the reasons for the empirical results. please read

The influence of high-speed rail on ice–snow tourism in northeastern China. Tourism Management（2020）, doi:10.1016/j.tourman.2019.104070.

Study on the Impact of High-speed Railway Opening on China's Accessibility Pattern and Spatial Equality[J].Sustainability 2018,10,2943. doi:10.3390/su10082943 .

The Impact of Accessibility on House Prices Based on a Geographically Weighted Regression Model[J]. Chinese Geographical Science，2018，28（3）：505-515. doi: 10.1007/s11769-018-0954-6

Reviewer #2: The authors used the data from Transportation Tomorrow Survey to explore the application of spatial availability and compare it to other relevant measures. The manuscript is valid and innovative. However, there also have some problems need to be solved.

1.In introduction, the authors only inserted the serial number of the references. The presentation is omitted. It is not standard enough.

2.There are many methods to measure the accessibility. What's the advantage and disadvantage of the improved methods compared with other methods, such as space syntax and potential model.

3.Some relevant references should be cited as follow:

Impact of Accessibility on Housing Prices in Dalian City of China Based on a Geographically Weighted Regression Model.Chin. Geogra. Sci. 2018 Vol. 28 No. 3 pp. 505–515 .https://doi.org/10.1007/s11769-018-0954-6

The influence of the spatial pattern of urban road networks on the quality of business environments: the case of Dalian

City. Environment, Development and Sustainability (2022) 24:9429–9446.https://doi.org/10.1007/s10668-021-01832-z

Differences in Accessibility of Public Health Facilities in Hierarchical Municipalities and the Spatial Pattern Characteristics of Their Services in Doumen District,China. Land 2021, 10, 1249. https://doi.org/10.3390/land10111249.

Transportation Accessibility Evaluation of Educational Institutions Conducting Field Environmental Education Activities in Ecological Protection Areas: A Case Study of Zhuhai City. Sustainability 2021, 13, 9392. https://doi.org/10.3390/su13169392.

6. PLOS authors have the option to publish the peer review history of their article (what does this mean?). If published, this will include your full peer review and any attached files.

Reviewer #1: No

Reviewer #2: No

---

## [Author Response · Author response to Decision Letter 0]

24 Oct 2022

Journal Requirements

• Thank you for this comment. We produced the manuscript using the .tex style template for PLOS ONE. 

2. Thank you for stating the following financial disclosure: "The Social Sciences and Humanities Research Council’s partnership grant: Mobilizing justice: towards evidence-based transportation equity policy." 

• Thank you for this comment. We’ve updated the statement accordingly and included it in the revised cover letter.

Important: If there are ethical or legal restrictions to sharing your data publicly, please explain these restrictions in detail. Please see our guidelines for more information on what we consider unacceptable restrictions to publicly sharing data: http://journals.plos.org/plosone/s/data-availability#loc-unacceptable-data-access-restrictions. Note that it is not acceptable for the authors to be the sole named individuals responsible for ensuring data access. We will update your Data Availability statement to reflect the information you provide in your cover letter.

• Thank you for this comment. The data set is available from https://soukhova.github.io/TTS2016R/ The full manuscript and the code are included in this repository: https://github.com/soukhova/Spatial-Availability-Measure. Both links are provided as footnotes in the Introduction section.

4. We note that Figures 3,4,5,7,8 and 9 in your submission contain copyrighted images. All PLOS content is published under the Creative Commons Attribution License (CC BY 4.0), which means that the manuscript, images, and Supporting Information files will be freely available online, and any third party is permitted to access, download, copy, distribute, and use these materials in any way, even commercially, with proper attribution. For more information, see our copyright guidelines: http://journals.plos.org/plosone/s/licenses-and-copyright. 

 We require you to either (1) present written permission from the copyright holder to publish these figures specifically under the CC BY 4.0 license, or (2) remove the figures from your submission. 

• Thank you for this comment. All images are original and were generated by us using the data sources stated in the manuscript using code found in the .Rmd file in the repository (https://github.com/soukhova/Spatial-Availability-Measure/blob/main/Spatial-Availability-Paper/Spatial-Availability-Paper.Rmd). The packages used to produce the figures are cited. 

 

Reviewer #1

The authors demonstrated how a widely used measure of accessibility with congestion fails to properly match the opportunity-seeking population.

1. Please highlight the key points in the abstract.

• Thanks for this comment. We modified the abstract to highlight the key points of this paper. The abstract now reads as follows:

Accessibility indicators are widely used in transportation, urban, and healthcare planning, among many other applications. These measures are weighted sums of reachable opportunities from a given origin conditional on the cost of movement, and are estimates of the potential for spatial interaction. Over time, various proposals have been forwarded to improve their interpretability: one of those methodological additions have been the introduction of competition. In this paper, we focus on competition, but first demonstrate how a widely used measure of accessibility with congestion fails to properly match the opportunity-seeking population. We then propose an alternative formulation of accessibility with competition, a measure we call _spatial availability_. This measure relies on proportional allocation balancing factors (friction of distance and population competition) that are equivalent to imposing a single constraint on conventional gravity-based accessibility. In other words, the proportional allocation of opportunities results in a _spatially available opportunities_ value which is assigned to each origin that, when all origin values are summed, equals the total number of opportunities in the region. We also demonstrate how Two-Stage Floating Catchment Area (2SFCA) methods are equivalent to spatial availability and can be reconceptualized as singly-constrained accessibility. To illustrate the application of spatial availability and compare it to other relevant measures, we use data from the 2016 Transportation Tomorrow Survey of the Greater Golden Horseshoe area in southern Ontario, Canada. Spatial availability is an important contribution since it clarifies the interpretation of accessibility with competition and paves the way for future applications in equity analysis. 

2. The introduction should further highlight the scientific problems, motivations, and possible innovations of the paper.

• Thank you for these points. We modified the text as follows to include the highlighted motivations (as a result of the scientific problems) at the end of the introduction. 

The key motivations of this paper are as follows:

- To address and improve on the interpretability of Hansen-type accessibility measure; and

- To consider competition from the perspective of the population for opportunities within an accessibility measure.

• We’d like to draw your attention to the summary of the objectives of the paper in the revised version of the manuscript and included as follows. We believe these correctly highlight the possible innovations of this paper.

In this way, the aim of the paper is three-fold:

- First, we aim to demonstrate that Shen-type (and thus @weibull_axiomatic_1976 accessibility and the popular 2SFCA methods) produce equivocal estimates of opportunities allocated as the result is presented as a rate (i.e., opportunities per capita);

- Second, we introduce a new measure, _spatial availability_, which we submit is a more interpretable alternative to Shen-type accessibility, since opportunities in the system are preserved and proportionally allocated to the population; and

- Third, we show how Shen-type accessibility (and 2SFCA methods) can be seen as measures of singly-constrained accessibility.

3. What is the contribution and why the contribution is important?

• An important question to ask indeed! We believe we highlight the contribution and its importance throughout the paper, particularly in subsection “Why does proportional allocation matter?” as well as in the conclusion. The contribution is in the conclusion is summarized as follows:

In this paper we show how a widely used measure of accessibility with competition (Shen-type accessibility) obscures some important internal values of opportunities taken. This is caused by confounding the population of zones with the _effective opportunity-seeking population_. We then propose an alternative derivation of accessibility with competition that we call spatial availability. This measure ensures that opportunities are allocated in a proportional way and preserved in the regional total. We also show that spatial availability and Shen-type accessibility are equifinal: formally the equations are the same (along with 2SFCA) and can be consider as singly-constrained measures. 

• In addition, following your suggestion, we edited the first few sentences of the conclusion’s second paragraph to make the importance of the contribution more clear. It now reads:

Spatial availability matters because competition is an important consideration for certain opportunity types and conventional Hansen-type accessibility does not capture it [@merlin2017competition]. Spatial availability also brings forward a different interpretation to competition than the Shen-type measure through its output value, the proportional allocation factors, and the output values per capita. In equity analysis and policy planning, an analyst might be interested in the internal values of their accessibility analysis, for example travel times, and who pays how much for accessibility. The increased interpretability and internal consistency of spatial availability can help to push accessibility analysis forward. Hansen-type measure tend to result in values which are very extreme as a result of multiple-counting opportunities as shown in empirical example. Multiple-counting may not be an issue if the opportunity-type is non-exclusive, but with the case of employment where one worker can only take one job, the resulting values are difficult to interpret (though it can be interpreted relatively to speak about urban form). In this paper, we also demonstrated how attempting to disentangle the absolute values of opportunities from the Shen-type measure is difficult as a result of Shen's definition which confounds the population with the effective-opportunity seeking population.

• We also added a sentence to the abstract which states the contribution of the paper more clearly, as inspired by this comment. The most current version of the abstract is included in reply to your comment 1.

4. The results should be better described, discussed and justified. I suggest the results should be better discussed and justified, such as whether they are consistent with previous studies or analyzing the reasons for the empirical results. please read:

• The influence of high-speed rail on ice–snow tourism in northeastern China. Tourism Management（2020）, doi:10.1016/j.tourman.2019.104070

• Study on the Impact of High-speed Railway Opening on China's Accessibility Pattern and Spatial Equality[J].Sustainability 2018,10,2943. doi:10.3390/su10082943

• The Impact of Accessibility on House Prices Based on a Geographically Weighted Regression Model[J]. Chinese Geographical Science，2018，28（3）：505-515. doi: 10.1007/s11769-018-0954-6

• Thank you for these references. They have been included in the revised version of the paper as examples of the important role that accessibility analysis plays in different disciplines. 

 

Reviewer #2

The authors used the data from Transportation Tomorrow Survey to explore the application of spatial availability and compare it to other relevant measures. The manuscript is valid and innovative. However, there also have some problems need to be solved. 

1. In introduction, the authors only inserted the serial number of the references. The presentation is omitted. It is not standard enough.

• Thank you for this comment. We are advised to use the in-text referencing style of PLoS ONE, which uses serial number of the reference.

2. There are many methods to measure the accessibility. What's the advantage and disadvantage of the improved methods compared with other methods, such as space syntax and potential model.

• Thank you for these points, discussing the advantages and disadvantages of similar methods is important to contextualize the improved method. In this paper, we do discuss other accessibility measures at length in the ‘Accessibility measures revisited’ section. We outline Hansen-type accessibility and then Shen-type accessibility, both of which are forms of potential accessibility. We introduce spatial availability in the following section and then discuss its advantages in the subsection ‘Why does proportional allocation matter?”.

• Our aim is to present a new accessibility measure, spatial availability, and we discuss its properties in reference to two comparable measures, Hansen-type accessibility, and Shen-type accessibility (2SFCA). These two are forms of potential accessibility (see for an example Zhang et al., 2021). We reviewed the space syntax model. As noted by Zhang et al. (2022, p. 9434). “[a] higher level of integration implies spatial clustering and better topological accessibility”. Some space syntax measures correlate well with accessibility, however accessibility is sufficiently distinct that it is not an appropriate comparator. 

3. Some relevant references should be cited as follow:

• The influence of the spatial pattern of urban road networks on the quality of business environments: the case of Dalian City. Environment, Development and Sustainability (2022) 24:9429–9446. https://doi.org/10.1007/s10668-021-01832-z

• Differences in Accessibility of Public Health Facilities in Hierarchical Municipalities and the Spatial Pattern Characteristics of Their Services in Doumen District, China. Land 2021, 10, 1249. https://doi.org/10.3390/land10111249.

• Transportation Accessibility Evaluation of Educational Institutions Conducting Field Environmental Education Activities in Ecological Protection Areas: A Case Study of Zhuhai City. Sustainability 2021, 13, 9392. https://doi.org/10.3390/su13169392

Thank you for these references, which we have added to the bibliography in the paper.

We trust that you will find that these revisions satisfy all the comments communicated. We are grateful for your insightful and constructive feedback, and appreciate that it helped to improve the paper.

---

## [Decision Letter · Decision Letter 1]

17 Nov 2022

Introducing spatial availability, a singly-constrained measure of competitive accessibility

PONE-D-22-24454R1

Dear Dr. Soukhov,

We’re pleased to inform you that your manuscript has been judged scientifically suitable for publication and will be formally accepted for publication once it meets all outstanding technical requirements.

Kind regards,

Jun Yang

Academic Editor

PLOS ONE

Additional Editor Comments (optional):

Accept

Reviewers' comments:

Reviewer's Responses to Questions

**Comments to the Author**

1. If the authors have adequately addressed your comments raised in a previous round of review and you feel that this manuscript is now acceptable for publication, you may indicate that here to bypass the “Comments to the Author” section, enter your conflict of interest statement in the “Confidential to Editor” section, and submit your "Accept" recommendation.

Reviewer #1: (No Response)

Reviewer #2: All comments have been addressed

2. Is the manuscript technically sound, and do the data support the conclusions?

Reviewer #1: (No Response)

Reviewer #2: Yes

3. Has the statistical analysis been performed appropriately and rigorously? 

Reviewer #1: (No Response)

Reviewer #2: Yes

4. Have the authors made all data underlying the findings in their manuscript fully available?

Reviewer #1: (No Response)

Reviewer #2: Yes

5. Is the manuscript presented in an intelligible fashion and written in standard English?

Reviewer #1: (No Response)

Reviewer #2: Yes

6. Review Comments to the Author

Reviewer #1: The authors have adequately addressed comments raised in a previous round of review and I feel that this manuscript is now acceptable for publication

Reviewer #2: All comments have been addressed. The authors have revised relevant problems. This paper could be accepted.

7. PLOS authors have the option to publish the peer review history of their article (what does this mean?). If published, this will include your full peer review and any attached files.

Reviewer #1: No

Reviewer #2: No

---

## [Editor Report · Acceptance letter]

11 Jan 2023

PONE-D-22-24454R1 

Introducing spatial availability, a singly-constrained measure of competitive accessibility 

Dear Dr. Soukhov:

I'm pleased to inform you that your manuscript has been deemed suitable for publication in PLOS ONE. Congratulations! Your manuscript is now with our production department. 

Kind regards, 

on behalf of

Dr. Jun Yang 

Academic Editor

PLOS ONE